# Spin relaxation of electron and hole polarons in ambipolar conjugated polymers

Remington L. Carey[1,6], Samuele Giannini [2,5,6], Sam Schott[1], Vincent Lemaur [2], Mingfei Xiao[1], Suryoday Prodhan[3], Linjun Wang [4], Michelangelo Bovoloni[2], Claudio Quarti [2], David Beljonne [2] & Henning Sirringhaus [1] ✉

The charge-transport properties of conjugated polymers have been studied extensively for opto-electronic device applications. Some polymer semiconductors not only support the ambipolar transport of electrons and holes, but do so with comparable carrier mobilities. This opens the possibility of gaining deeper insight into the charge-transport physics of these complex materials via comparison between electron and hole dynamics while keeping other factors, such as polymer microstructure, equal. Here, we use field-induced electron spin resonance spectroscopy to compare the spin relaxation behavior of electron and hole polarons in three ambipolar conjugated polymers. Our experiments show unique relaxation regimes as a function of temperature for electrons and holes, whereby at lower temperatures electrons relax slower than holes, but at higher temperatures, in the so-called spin-shuttling regime, the trend is reversed. On the basis of theoretical simulations, we attribute this to differences in the delocalization of electron and hole wavefunctions and show that spin relaxation in the spin shuttling regimes provides a sensitive probe of the intimate coupling between charge and structural dynamics.

One of the characteristics of a clean, ambipolar semiconductor is the ability to support with comparable carrier mobilities the transport of both electrons in the conduction band and holes in the valence band. Any differences between the electron and hole mobilities should reflect differences in the intrinsic electronic structure of the conduction and valence band states, but should not be the consequence of defect states in the band gap that might trap one of the two carriers more strongly than the other. Of course, unipolar semiconductors, in which one type of charge is significantly more mobile than the other, can be very useful for many applications, but ambipolar semiconductors are particularly interesting for fundamental studies because they allow one to compare the charge-transport properties of carriers in the valence and conduction bands. Stimulated by the discovery that mobile electron transport can be observed in conjugated-polymer field-effect transistors (FETs) when specific electron-trapping groups associated with silicon dioxide gate dielectrics are avoided[1], research into ambipolar conjugated polymers has grown steadily over the past two decades[2–4] and has enabled a number of unique device structures. Examples include complementary-like logic circuits based on ambipolar inverters—for which organics offer simpler fabrication procedures—and light-emitting field-effect transistors, in which the position of the recombination zone at the boundary between electron and hole accumulation layers in the channel can be controlled via applied voltage[5].

Comparing the properties of electron and hole carriers in ambipolar polymers also provides unique insights into the charge-transport physics of these complex material systems[6]. The utility of such experiments is that they provide insight into how differences in the

[1]Cavendish Laboratory, University of Cambridge, Cambridge CB3 0HE, UK. [2]Laboratory for Chemistry of Novel Materials, University of Mons, 7000 Mons, Belgium. [3]Department of Chemistry, University of Liverpool, Liverpool L69 3BX, UK. [4]Key Laboratory of Excited-State Materials of Zhejiang Province, Department of Chemistry, Zhejiang University, Hangzhou 310058, China. [5]Present address: Institute of Chemistry of OrganoMetallic Compounds, National Research Council (ICCOM-CNR), I-56124 Pisa, Italy. [6]These authors contributed equally: Remington L. Carey, Samuele Giannini. ✉e-mail: hs220@cam.ac.uk

electronic structure experienced by electrons and holes affect charge transport properties without changing other important factors, such as the polymer microstructure or vibrational dynamics, that are impossible to keep unchanged when comparing different polymer systems.

Field-induced electron spin resonance (FI-ESR) spectroscopy is a particularly powerful experimental technique for studying the charge-transport physics of organic semiconductors and has been used to estimate the spatial extent of polaron wavefunctions[7,8], observe polaronic motional narrowing[9–12], and study charge-trapping mechanisms in organic FETs (OFETs)[8,11,13]. Though the technique has not yet been applied to the aforementioned intrinsically ambipolar OFETs based on single-component conjugated polymers, it has been used to study blend-based ambipolar OFETs comprising a mixture of a p-type π-conjugated polymers with n-type fullerenes[2,14,15]. In these composite materials, it was long-suspected that positive carriers were injected into the polymer and negative ones into the fullerene. A definitive proof came from the studies of Marumoto et al., who used used FI-ESR to identify unique g-values in the positive and negative bias regimes[16]. Watanabe et al. added to this work and investigated the role of the gate electrode in the ambipolar charge-transport characteristics of such devices[17].

Recently, our group used FI-ESR to systematically study the spin dynamics of hole polarons in several high-mobility organic semiconductors[18]. By performing power-saturation measurements from 5 K to room temperature, we were able to extract the spin-lattice/longitudinal and spin-spin/transverse relaxation times, $T_1$ and $T_2$. The former corresponds to the decay of the paramagnetic magnetization of the sample (induced by the external magnetic field) along the quantization axis and is caused by spins flipping by exchanging energy with the lattice. The latter corresponds to decay along the transverse direction and can be caused by energy-exchanging or energy-conserving processes; one pertinent example is the decoherence of two spins placed in different local magnetic environments. The paramagnetic species relevant to FI-ESR are the polarons induced in the accumulation layer upon application of a gate voltage.

In our previous work[18], we observed three regimes of spin relaxation: *inhomogeneous broadening* at low temperatures, where local (spatial) variations in the hyperfine and spin-orbit coupling fields result in a Gaussian spread of Lorentzian lineshapes; *motional narrowing* at intermediate temperatures, in which spins move quickly enough such that they all experience the same local environment on average and thus narrow the resonance signal; and a high-temperature relaxation regime that had not previously been observed and in which the lineshape again broadens despite the increased hopping frequency of polarons. Of these regimes, the first two are predicted by the Redfield theory of spin relaxation[19], while the third requires an alternate explanation. By excluding inter-chain effects and simulating intra-chain dynamics of the polaronic wavefunction on picosecond time-scales, we hypothesized a *spin-shuttling* model of relaxation in this regime, whereby reconfigurations of the wavefunction in response to nuclear vibrations cause spins to relax. Unfortunately, we were not able to test this dependence on wavefunction dynamics at that time because our study[18] relied on comparisons between polymers, but such comparisons also naturally include variation in system microstructure and vibrational dynamics, both of which have confounding effects on spin relaxation.

To probe this regime more directly and to obtain a deeper understanding of all three relaxation regimes, here we apply the FI-ESR technique to three ambipolar conjugated polymers (described in detail below) and systematically compare the spin relaxation dynamics of electron and hole polarons in the same, single-component polymers from 5 K to room temperature. Because the microstructure and vibrational dynamics experienced by the two types of carriers are identical, we are able to identify the role played uniquely by the

polaronic mobility and shape and distribution of the carrier wavefunctions along the polymer backbone. We show that the three regimes of spin relaxation are present for both types of polarons in all three systems. By relating charge-transport measurements to FI-ESR spectra, we show that polaron spin dynamics at temperatures below 150 K are driven primarily by charge motion and are largely independent of wavefunction localization. For higher temperatures, we show that wavefunction extent becomes the dominating factor, with spin-shuttling driven by nuclear torsion relaxing spins faster than charge transport can counteract this effect. We compare these observations to quantum chemical calculations on DPPT-TT, which show a clear difference in the localization of holes and electrons. We use this to identify in a DPPT-TT chain the exact site of torsion responsible for spin-shuttling, and we compare these results to those of the fused systems, where the expected similarity of the wavefunctions manifests itself in a more similar relaxation behavior for electrons and holes.

## Results

### Material systems

We consider three representative ambipolar polymer systems here, the chemical structures of which are shown in the top row of Fig. 1. First is the diketopyrrolopyrrole (DPP) derivative DPPT-TT[20], which is a donor-acceptor system comprising alternating electron-deficient DPP and electron-rich thiophene-thieno[3,2-b]thiophene (TT) moieties along the polymer backbone[21]. Chen et al.[22] demonstrated ambipolar transport in DPPT-TT under optimized device fabrication. In particular, high annealing temperatures (>200 °C) improved edge-on packing and facilitated charge transport for both holes and electrons, while solely electron transport was improved by even higher annealing temperatures (320 °C). Injection of electrons was improved by omitting the common plasma-cleaning step of the gold contacts before spin-coating (which lowers the electrode work function). Mobilities on the order of 1 cm²V⁻¹s⁻¹ were achieved for both polaron types.

The other two polymers—an anthracene-naphthalene derivative (AN) and a naphthalene-naphthalene derivative (NN)—are electron-deficient, low-band-gap conjugated polymers whose backbone structures do not contain any single bond linkages[23]; both have been previously characterized by Xiao et al. (2021). (In Xiao et al. work, note that AN here is referred to as AN2 there and NN here as NN1 there.)[24] In these systems, the double bond linkages between the fused conjugated units result in a highly rigid structure and rod-like conformation. Though the structures are rigid, the backbones are not entirely planar due to steric repulsion between carbonyl oxygens and adjacent C-H groups, which results in both systems having a torsion angle of roughly 18° at the double bond. It is important to note that in these polymers, all conjugated units are electron deficient, i.e., these are not donor-acceptor polymers (in contrast to DPPT-TT). Though ambipolar transport was demonstrated for these polymers, only the electron regime was systematically studied due to injection issues for holes. In AN, the electron mobility ranged from 0.2 to 0.6 cm²/Vs (depending on molecular weight), while in NN the highest mobility achieved was 0.1 cm²V⁻¹s⁻¹ for high molecular weights, and was drastically lower for smaller molecular weights.

### Device characterization

Figure 1 shows the n- and p-type transfer curves and corresponding mobilities in all three systems measured at 290 K. As shown in Fig. 1d, the DPPT-TT device performance is nearly identical for both holes and electrons, each having similar threshold voltages and turn-on. In Fig. 1e, f, we see that the n-type transfer characteristics of the AN and NN polymers exhibit a lower threshold voltage, more rapid turn-on, and higher ON-current, while the p-type exhibits a higher threshold voltage and more gradual turn-on. The bottom row of Fig. 1 shows the extracted mobilities of the three systems. In AN at 290 K (Fig. 1h), both electron and hole mobility are on the order of 0.1 cm²/Vs, which is

lower than the electron value of 0.4–0.8 cm²/Vs in the fully optimized devices reported by ref. 24 (Mobilities for holes were not reported.) In NN (Fig. 1i), both polaron mobilities were about 0.01 cm²/Vs, which is the expected value for electron mobility[24]. In DPPT-TT (Fig. 1g), our observed mobilities are also lower than those reported in Chen's work[22]: 0.1 vs. 1 cm²/Vs for both holes and electrons. The difference in mobility to the values reported in literature is most likely due to the fact that for the FI-ESR measurement we need to use an elongated, large-area device architecture that fits into the ESR cavity, for which it is difficult to fully optimize the polymer microstructure and processing conditions[18]. However, we emphasize that for all three polymers we are able to operate the devices in a regime in which electron and hole transport are reasonably well-balanced, with the n-type mobility being less than a factor of 2–3 higher than the p-type mobility for a given magnitude of gate voltage.

The mobilities shown in the bottom row of Fig. 1 exhibit a dependence on gate voltage and somewhat different onset voltages. In principle, differences in the onset voltages of electrons and holes should be taken into account when comparing mobilities and spin relaxation times. However, despite the gate-voltage-dependent mobilities, the device characteristics of our ambipolar devices (in particular DPPT-TT and AN) are very well-behaved and the onset voltages are very close,

typically between ±5 V and 0 V for both polaron types. Since we measured the ESR spectra at high magnitudes of gate voltage (±60 V), the associated uncertainty in the mobility is estimated to be less than 10%. This uncertainty is much smaller than the difference between electron and hole mobility at a fixed magnitude of gate voltage (which is a factor of approximately 2 in the case of DPPT-TT). Therefore, given that the origin of the relatively small variations in onset voltage is not known, we compared electron and hole ESR spectra at the same gate voltage, i.e., total induced charge concentration, and did not correct for the small differences in onset voltage.

Spin relaxation times were recorded via the following procedure: At a given temperature, we first measured the ESR spectra in the p-type regime at a sufficiently high gate voltage ($V_g = -60$ V) as a function of microwave power. We then measured the background signal ($V_g = 0$ V) if deemed necessary from initial test scans—see Supplementary Note 1, then measured the n-type spectra ($V_g = 60$ V) and corresponding background signal (if necessary) before moving on to the next temperature. Transfer and output curves were recorded before and after each ESR measurement to detect any device degradation or malfunction over the scanning time. In order to avoid systematic error in ESR curves due to this p-type, background, then n-type measurement sequence, we occasionally and randomly measured n-type data first or

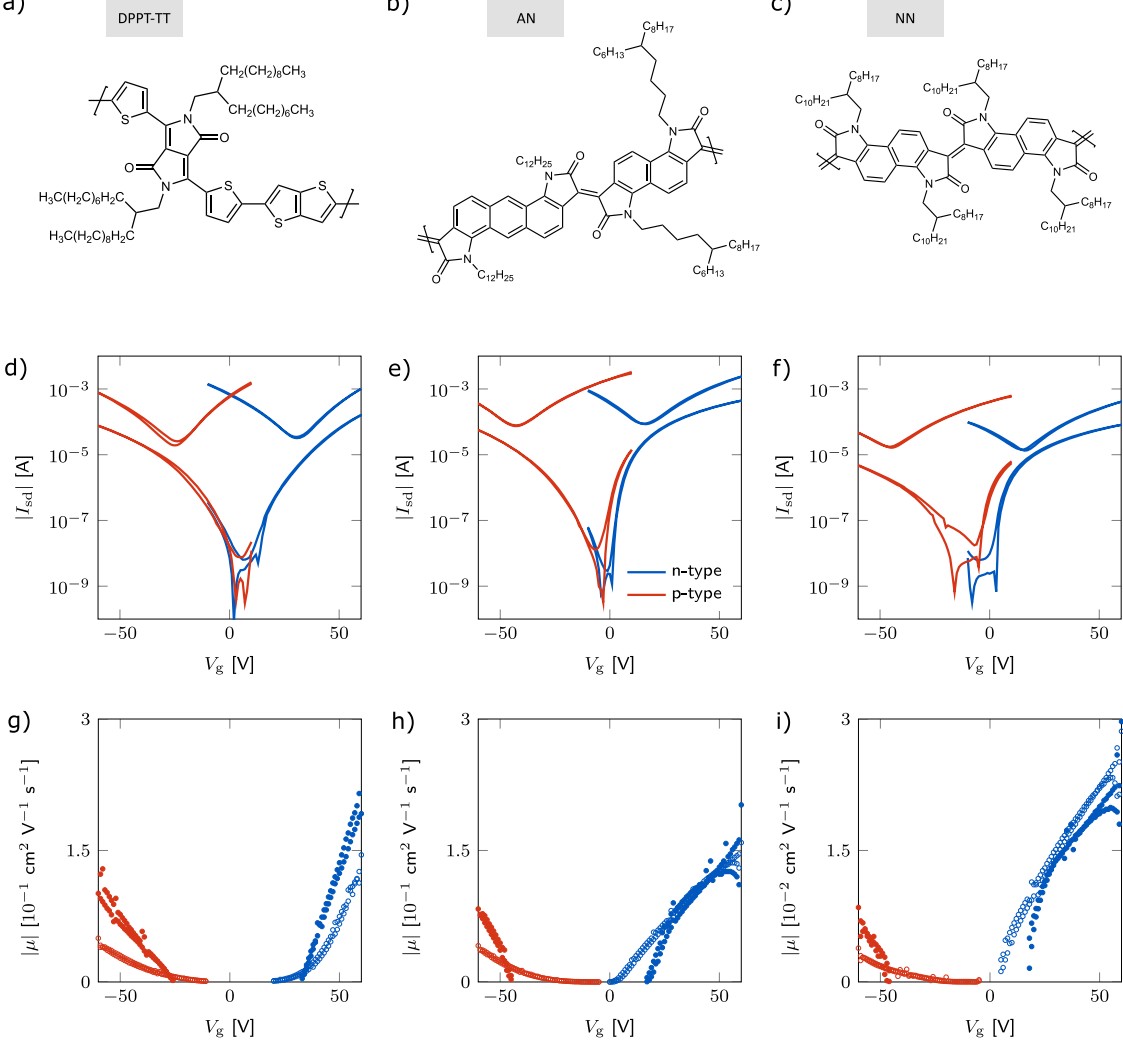

**Fig. 1 | Ambipolar FET characteristics for DPPT-TT, AN, and NN at room temperature. a–c** The chemical structures of the three molecules used for this study: DPPT-TT, AN, and NN, respectively. **d–f** Device transfer curves in both the linear and saturation regimes. Red corresponds to holes and blue to electrons. In all plots, the upper, v-shaped trace corresponds to the saturation regime (as expected for ambipolar performance). **g–i** Corresponding mobilities (where, for clarity, mobilities before the turn-on voltage are not shown). The open (filled) symbols correspond to the linear (saturation) mobilities.

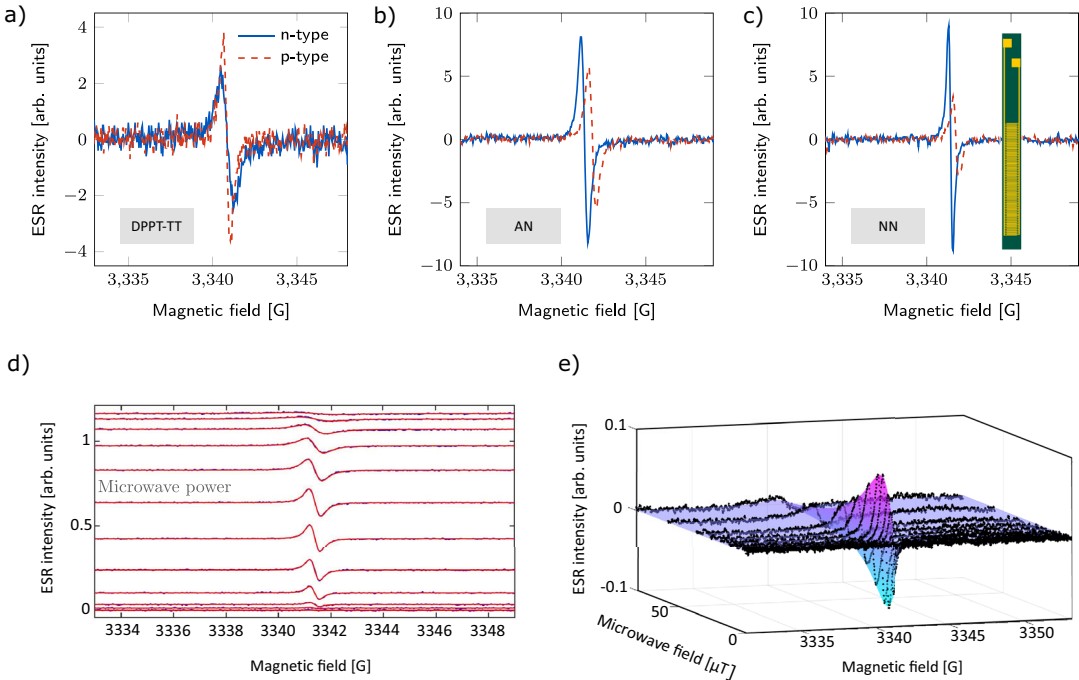

**Fig. 2 | Representative ambipolar FI-ESR spectra. a–c** Fixed-power spectra for holes and electrons in DPPT-TT, AN, and NN, respectively, at 200 K. The solid blue trace corresponds to electrons and the dashed red trace to holes. (1c inset): A schematic of a field-induced ESR sample, showing the source, drain, and inter-digitated gate electrodes. **d** Stacked plots of the n-type spectra in AN at 200 K as a function of microwave power. The block dots are data points while the red traces are fits to the data. **e** The same spectra as in (**d**) presented as a surface plot with the full 2D fit. ESR spectra at 10 and 100 K are shown in Supplementary Note 3 for comparison.

measured n-type performance in another device to compare. No differences were observed.

For DPPT-TT and AN at temperatures below 80 K, hole injection was so low that an ESR signal was not obtainable. (The minimum number of spins required is $10^{11}$.) To overcome this, we recorded those spectra by injecting polarons at 150 K and then lowering the temperature to the required value to measure. Consequently, FET curves are not available for these low temperatures, as recording these would have emptied the accumulation layer of injected polarons. For NN this problem occurred below 120 K, so polarons were injected at 200 K. It is worth noting that we were still able to inject electrons below these hole-limiting temperatures, at least for another 30-50 K. However, because this research relies on comparisons between holes and electrons within the same system, we decided that the best procedure would be to inject both polaron types at the same temperature for these specific data points.

**Relaxation dynamics**

To extract relaxation times, power-saturation curves were recorded at each temperature step. A mathematical summary of the method is given in Supplementary Note 2, while our previous work describes it in more detail[18]. In essence, power-saturation measurements probe the ESR signal as a function of both magnetic field and microwave power. In a standard ESR experiment, where only the magnetic field is swept, the width of the resonance signal is determined by the spread of different resonance positions among the spin ensemble, a quantity characterized by $T_2$; these measurements, therefore, give an estimate for the transverse relaxation times. Power-saturation measurements examine how the signal evolves with increasing microwave power. At high powers, the number of spin-flip-inducing photons per unit time is so high that the spin ensemble reaches equilibrium (having an equal number of spins in the two states) and is unable to relax between field scans; this results in no signal observed at resonance since the two populations are equal. Because relaxation-from-equilibrium is

described by $T_1$, these experiments provide an estimate for the longitudinal relaxation time. We, therefore, fit the full 2D dependence of the resonance signal on field position and microwave power to extract $T_1$ and $T_2$ unambiguously.

Representative ESR spectra are shown for each system at 200 K in the top row of Fig. 2. For all three systems, distinct electron and hole signals were observed for positive and negative voltages, respectively. In DPPT-TT (Fig. 2a) the two g-factors are the same within measurement error, $g_e = g_h = 2.00385$, while in AN and NN (Fig. 2b, c) the resonance positions are slightly different: $g_e = 2.00367$ and $g_h = 2.00335$ in AN, and $g_e = 2.00367$ and $g_h = 2.00340$ in NN (All values are reported with error ± 0.00005). It is important to note that the relative spectral positions and intensities are not directly comparable between the different polymers since these vary with measurement parameters that may change between measurements, such as the exact position of the sample within the cavity and the resonance frequency of the spectrometer. In Fig. 2d, a representative stacked plot depicting the evolution of the electron polaron spin signal in AN at 200 K as a function of microwave power is shown; the black dots are individual data points while the red lines are fits to extract relaxation times. To the right of this, in Fig. 2e, is the same data depicted as a surface plot along with the full 2D fit.

The full temperature dependencies of the longitudinal and transverse relaxation times extracted from these fits can be found in Supplementary Note 4. In accordance with ref. 18, the inhomogenous broadening, motional narrowing, and spin-shuttling regimes of relaxation are clearly visible in all three systems for both polaron types. Because $T_1$ monotonically decreases with increasing temperature, only the behavior of $T_2$ distinguishes between the relaxation regimes. We, therefore, show only the behavior of $T_2$ in these systems in Fig. 3a–c. In the three figures, the different regimes are delineated by red and blue arrows (for electrons and holes, respectively), of which there are two (per polaron) in each plot: one to mark the transition from inhomogenous broadening to motional narrowing, and one for motional

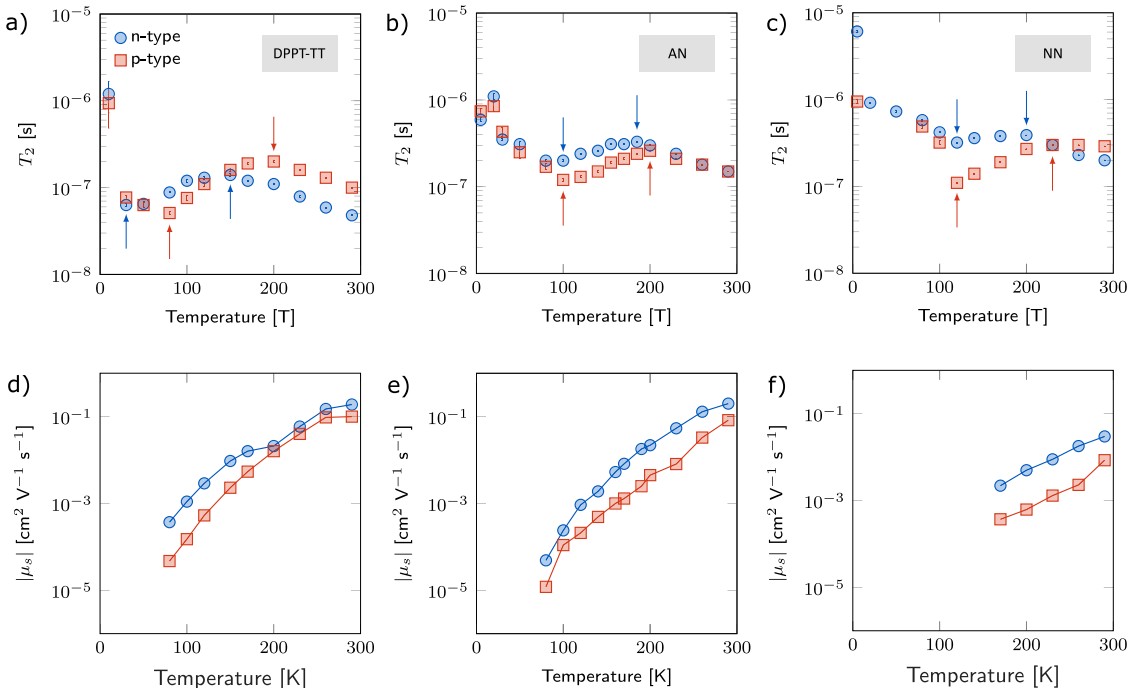

**Fig. 3 | Extracted transverse relaxation times ($T_2$) and saturation mobilities from 5 to 290 K. a–c** Dependence on temperature of the n- and p-type polaronic transverse relaxation times in DPPT-TT, AN, and NN, respectively. The red and blue arrows mark a transition in relaxation regime for holes and electrons, respectively. The first (per color per plot) marks the transition from inhomogenous broadening to motional narrowing, and the second from motional narrowing to high-temperature relaxation. The n-type $T_2$ data for AN and NN has previously been published[24], but is shown here for comparison. **d–f** Dependence of polaron mobilities on temperature across the three different materials. At low temperatures, polaronic mobilities differ by over an order of magnitude in DPPT-TT, while they are within an order of magnitude in AN (Low-temperature data is not available for NN for reasons mentioned in the main text). Above 200 K, polaronic mobilities are nearly equal in DPPT-TT, while in AN and NN there remains a significant difference.

narrowing to high-temperature relaxation. In the fused systems, the transition from inhomogenous broadening to motional narrowing occurs nearly at the same temperature for both hole and electron polarons: at 100 K in AN and at 120 K in NN. High-temperature relaxation, on the other hand, sets in slightly later for holes than for electrons: in AN, electrons transition at 180 K while holes transition at 200 K, and in NN electrons transition at 200 K while holes transition at 220 K. In DPPT-TT motional narrowing sets in much earlier—at about 40 K for electrons and 80 K for holes—while high-temperature relaxation sets in at about 150 K for electrons and at 200 K for holes. Interestingly, while in the motional narrowing regime spin relaxation is faster for holes than for electrons in all three polymers, this is reversed in the spin-shuttling regime in NN and DPPT-TT, and in AN the polaronic lifetimes are roughly equal. The n-type $T_2$ data for AN and NN has previously been published[24], but is reported here for the purposes of comparison to p-type performance and to DPPT-TT.

To interpret these results, it is necessary to examine both the charge-carrier mobility and delocalization of polaronic wavefunctions. The charge-carrier mobility extracted from the FET transfer characteristics is plotted in the bottom row of Fig. 3. We see that for all systems at all temperatures, electron mobility is higher than that of holes. This difference is about an order of magnitude in the fused systems. In DPPT-TT the mobility difference is also about one order of magnitude below 200 K, but narrows at higher temperatures. At room temperature electron mobilities for all systems are about 0.1 cm² V⁻¹ s⁻¹, and drop four orders of magnitude as the temperature is lowered to 80 K. Due to aforementioned injection issues, FET curves below 80 K in AN and DPPTT-TT and below 150 K in NN were not recorded.

The delocalization of the electron and hole polaron wavefunctions can, in principle, be investigated experimentally by analyzing the hyperfine broadening of the ESR signal at low temperatures[19]: If a Gaussian signal is observed in the inhomogeneous broadening regime

at the lowest temperatures, then its root-mean-square width is determined by the variance of local fields experienced by individual spins in the system. By extracting this parameter from fits, the average variance and thus average extent of the wavefunction can be calculated. This typically requires measuring at 5 K, where electrons are frozen in their local environments. Unfortunately, this method was not possible for the data presented here, as all curves remained Lorentzian even down to 5 K. Thus, the hyperfine coupling had to be calculated by alternate means.

Transverse demagnetization can be caused by pure decoherence due to differing magnetic environments or by spin-flips, meaning $T_1$ contributes partially to $T_2$. Removing this contribution yields the decoherence time, $T_2^*$, which couples directly to the root-mean-square fluctuation of the local magnetic fields, $B_{\text{rms}}$[18]. Decoherence in the inhomogenous broadening regime is described by

$$T_2^* = \tau_c + \frac{1}{\gamma_e B_{\text{rms}}}\left(1 + \frac{1}{\gamma_e B_{\text{rms}}\tau_c}\right), \qquad (1)$$

and in the motional narrowing regime by

$$T_2^* = \frac{1}{\gamma_e^2 B_{\text{rms}}^2 \tau_c}, \qquad (2)$$

where $\tau_c$ is the characteristic timescale on local fields fluctuate and $\gamma_e$ is the free electron gyromagnetic ratio. In organic semiconductors, it is reasonable to equate $\tau_c$ to the inverse of charge motion frequency and to take the local field fluctuations as resulting from local hyperfine fields[18]. Then, the crossover from inhomogeneous broadening to motional narrowing results from a shortening of $\tau_c$ (which is an increase in the motion frequency) due to the temperature-activated motion of charges. At this crossover point, both expressions must

**Table 1 | Hyperfine coupling parameters**

| | AN | | NN | | DPPT-TT | |
|---|---|---|---|---|---|---|
| | *n* | *p* | *n* | *p* | *n* | *p* |
| Crossover temperature [K] | 100 | 100 | 120 | 120 | 30 | 80 |
| Corresponding $T_2^*$ [ns] | 204 | 125 | 339 | 108 | 63.3 | 52.0 |
| Extracted $B_{rms}$ [G] | 0.84 | 1.40 | 0.54 | 1.62 | 2.70 | 3.28 |

Crossover temperature, corresponding decoherence time ($T_2^*$), and extracted $B_{rms}$ at the point of crossover from inhomogeneous broadening to motional narrowing.

accurately give $T_2^*$, thus allowing one to combine them to calculate $B_{rms}$. This procedure was used for both polaron types in the three polymers here. The calculated values are shown in Table 1. We see there that in all three systems holes have a larger $B_{rms}$ than electrons. Because $B_{rms}$ scales as $1/\sqrt{N}$, where $N$ is the number of nuclei over which a wavefunction is spread, this indicates that holes are more localized than electrons in each of the three polymers studied. As will be discussed below, this conclusion is supported by quantum-chemical calculations in DPPT-TT. Furthermore, we note that the relatively small $B_{rms}$ values in AN and NN may be due to the fact that polarons are able to delocalize over relatively large distances in these fused systems[23]. At present this is a hypothesis, as direct polaron localization lengths for AN and NN have yet to be calculated. Finally, it is worth noting that the only homopolymer, NN, has the largest difference between electron and hole hyperfine field strength (0.54 G vs. 1.62 G). This was unexpected and additional simulations are likely required to understand this.

With mobility and hyperfine data presented, we are now in a position to interpret the relaxation times displayed in Fig. 3. The differences of the onsets of the motional narrowing regime for electrons and holes can be understood well in terms of the measured mobility values. In AN, the almost equal onset of motional narrowing for polarons is explained by very similar mobilities of electrons and holes around the crossover temperature. Below 100 K, both electrons and holes hop relatively slowly, meaning they don't experience enough hyperfine field fluctuations (per unit time) to average-out local field differences. Only after 100 K are their mobilities high enough to ensure a motionally narrowed resonance. (In NN, we cannot perform this analysis as we were unable to measure device mobilities at the low temperatures where motional narrowing sets in.) In contrast, for DPPT-TT, there is over an order-of-magnitude difference in polaron mobilities at 100 K, and this difference increases with decreasing temperature. Thus, because electrons are so much more mobile than holes in this regime, their motion is rapid enough to average-out local differences at much lower temperatures than holes. Generally, the fact that spin relaxation of electrons is slower than that of holes in the motional narrowing regime for the polymers here is fully consistent with the fact that electron mobility is higher than hole mobility in all three systems.

We now turn to a discussion of our key observation that the spin-shuttling regime sets in earlier for electrons and that electrons tend to be more strongly affected by the spin-shuttling relaxation mechanism than holes. Clearly, in this regime spin relaxation is not correlated with FET mobility, unlike in the motional narrowing regime. As originally proposed by ref. 18, in the spin-shuttling regime the coupling of charges to active molecular vibrational modes—such as torsional vibrations—shuttles the electronic wavefunctions back-and-forth along the polymer backbone many times before a hopping event, and each shuttling event presents an opportunity for relaxation. We expected in this regime that relaxation times would not directly reflect differences in carrier mobilities (as would be predicted in the motional narrowing regime). Instead, because this regime is governed by the coupling of spins to the vibrational, structural dynamics of the chain, we expected

that differences in the spatial location and extent of carrier wavefunctions would be critical and spin relaxation would be determined by whether polaron wavefunctions have significant amplitude around areas of the backbone where torsional vibrations occur. In other words, in this framework spins are less strongly affected by the spin-shuttling relaxation mechanism if their wavefunctions are localized in regions of the chains that are located away from relevant sites of torsion.

Finally, it is important to acknowledge that, in principle, mobility can be severely limited by specific defect configurations that are not representative of the spin environment probed by ESR experiments. However, in the relatively high-mobility polymer systems we are investigating here, we observe generally a good correlation between the motional narrowing physics observed in ESR and the measured device mobilities—a trend also shown in our previous work for other high-mobility polymer systems[18]. Also, in the present work, the observed correlation in the motional narrowing regime between the more effective spin relaxation of electrons and the higher electron mobility is evidence that transport in these systems occurs in a regime in which the configurations probed by ESR are representative of the environments that determine the charge mobility. This is probably a consequence of the relatively high mobility of the polymers investigated, where specific, "minor" defect sites acting as deep charge traps do not play an important role.

## Simulations on DPPT-TT

To interpret our experimental data, we calculated the electronic structure of DPPT-TT and used state-of-the-art numerical propagation of the hole and electron carrier wavefunctions coupled to nuclear motion. These calculations consider transport of polarons along single polymer chains and do not consider hopping between chains. This is deliberate because our earlier work[18] found the spin-shuttling-induced spin relaxation to be an intra-chain phenomenon. We selected DPPT-TT for this detailed theoretical analysis because it provided the highest charge-carrier mobilities and had the largest difference between electron and hole relaxation times in the spin-shuttling regime. To describe the electronic structure of DPPT-TT we used a Su-Schrieffer-Heeger (SSH)-type model Hamiltonian[25], which considers both local and non-local electron-phonon (e-ph) couplings. In particular, as described in Supplementary Note 6, the dynamics of the nuclei are described by three effective, harmonic vibrational modes. One is a high-frequency ($\approx 1000$ cm$^{-1}$) intra-monomer mode, while the other two are low frequency vibrations (<40 cm$^{-1}$) coupled to an "external" mode (intended as an additional contribution from the environment to the reorganization energy, as further described below), and to an inter-monomer torsional mode between successive monomer units along the polymer axis. It is worth noting that these soft degrees of freedom are active even at low temperature and, in principle, they also tend to reduce spin lifetimes at lower temperature, but in a less effective manner than at high temperature. We note in passing that the transition between motional narrowing and spin shuttling should not be interpreted as a transition at which suddenly a particular vibrational mode starts to be activated that leads to spin shuttling, but rather as the temperature at which the spin relaxation from motional narrowing becomes so weak that the relaxation induced by spin-shuttling becomes dominant. Our SSH Hamiltonian (eq. 7 in Supplementary Note 6) was parameterized by using ab-initio periodic DFT calculations to incorporate important band-structure properties of the (infinite isolated) polymer chain, and quantum chemical calculations on shorter polymer chains and individual fragments to compute charge distribution, relaxation energies and electron-phonon coupling values. Details of the calculations and related parameterization can be found in Supplementary Notes 5–9.

Our periodic DFT calculations show that intra-chain electronic couplings and bandwidths are larger for electrons than for holes (see

Supplementary Fig. 5a and Supplementary Table 1). To investigate the impact of the T-TT torsion angle (which separates the donor and acceptor units) on electronic structure in more detail, we calculated the electronic couplings between the frontier orbitals as a function of increasing torsion. Supplementary Fig. 5b shows that from 0 to 90° electrons have a higher electronic coupling than holes, although this difference is greatest at small angles and decreases as $\theta_{\text{T-TT}}$ increases; as expected, both values decrease as the torsional angle increases. These calculations show that the coupling of the electronic structure to torsion is slightly stronger for electrons than for holes. We also calculated the relaxation energies upon oxidation and reduction of DPPT and TT units (see Supplementary Table 2 and Supplementary Fig. 6). Although relaxation energies are similar for the excess hole and electron in the TT monomer, we found that the hole relaxation energy is twice as large as the electron relaxation energy in the DPPT monomer. Notably, the DPPT-TT polymer chains behave like quantum wells, with shallower HOMO and LUMO energy levels located on the DPPT part (see Supplementary Table 1). As a result, we expect the motion of the thermalized charge carriers—either holes or electrons—to be more strongly affected by local electron-phonon interactions on the DPPT units.

The observations above are consistent with the fact that experimentally electrons are more mobile than holes. To quantify the (temperature-dependent) hole and electron carrier mobilities in single conjugated DPPT-TT chains and the related wavefunction delocalization, we used mixed quantum-classical dynamics (see Methods for details). In particular, we employed the advanced crossing-corrected variant[26,27] of Tully's fewest switches surface hopping algorithm[28] to propagate the electronic wavefunction $|\Psi(t)\rangle$ (eq. (3)) of the system in time, thereby giving access to electronic transport properties and a first-principles view of the charge-carrier motion. In this work, we included fundamental local and non-local electron-phonon interactions and we parameterized our model as detailed in the Supplementary Information. The time-evolved wavefunctions were calculated from a swarm of individual trajectories, which is meant to reproduce the dynamics of quantum wavepackets. These trajectories were propagated for 1 ps to reach the equilibrium diffusion regime and then averaged to compute an ensemble charge density and the mean squared displacement (MSD) in eq. (6). Next, we extracted hole and electron mobilities (eq. (8)) after calculating the diffusion coefficient and found them to be approximately 1–2 and 9–20 cm²V⁻¹s⁻¹, respectively, at 300 K (depending on the amount of external relaxation energy, as discussed below). This predicted difference between electron and hole mobilities is qualitatively consistent with our measurements here and with the mobility values reported by ref. 22 for DPPT-TT in optimized FET devices, i.e., 1.36 cm²V⁻¹s⁻¹ for holes and 1.56 cm²V⁻¹s⁻¹ for electrons. However, the predicted mobilities, especially for electrons, are overestimated most likely because we only consider transport along single polymer chains and do not consider the hopping between chains.

We next computed mobilities as a function of temperature and compared them to the experimentally observed temperature dependence presented in Fig. 1d and reproduced in Fig. 4 for convenience. The computed mobilities were evaluated between 150 and 300 K, where nuclear quantum effects are expected to be relatively small. We see that computed mobilities show the same thermally activated trends as observed in our FI-ESR measurements, though, as discussed above, the calculated values are up to one and two orders of magnitude larger than experimental mobilities for holes and electrons, respectively, and the theoretical temperature dependence is weaker. Part of this discrepancy can be explained by the fact that our simulations only consider local electron-phonon coupling to the usual high-frequency aromatic breathing mode, neglecting slow modulations in site energies associated with fluctuations in the local dielectric environment. Including an additional, effective, low-frequency vibration

with associated relaxation energy $\lambda_{ext} = 100$–150 meV indeed brings calculated mobilities closer to the experimental values (see Fig. 4). We hypothesize that the remaining difference between theory and experiment is due to (1) inter-chain motion and trapping defects being omitted in our model, and (2) the DPPT-TT microstructure not fully being optimized in the present device measurements (which also explains the smaller mobilities measured here compared to those reported in by ref. 22). Indeed, the general understanding of the transport physics of these relatively high-mobility conjugated polymers is that device mobility is limited by the slow interchain hopping processes that charges undergo when they come to the end of a chain or when they encounter a conjugation defect along the polymer backbone; these occur on a length scale on the order of 10 nm. The microstructure of these high-mobility systems is sufficiently uniform that it is not specific, minor defect sites and deep trap states that limit mobilities, but rather these typical interchain hopping processes that need to occur on a 10 nm length scale. This is ultimately the reason we observe such a clear correlation between spin relaxation times in the motional narrowing regime and experimentally observed mobilities. As shown in our previous paper[18], it is possible to use the experimentally determined motion frequencies in the motional narrowing regime to estimate the corresponding hopping distances by using the Einstein relationship. If we perform this analysis on the present DPPT-TT system, the estimated hopping distance at 150 K is on the order of 15 nm, consistent with the interpretation that it is a regular interchain hopping process and not a minor defect site that governs the mobilities and spin lifetimes in the motional narrowing regime.

Despite the above, it is important to note that our simulations are consistent with our FI-ESR measurements: computed charge-carrier mobilities are larger for electrons than for holes. To investigate the reason for such a finding we computed the inverse participation ratio (IPR) (eq. (9)), which is a proxy for the intra-chain charge delocalization, and calculated an average < IPR > over time and trajectories (indicated with < ... >). This analysis suggests that the photogenerated electron at 300 K is delocalized on average slightly beyond a DPPT-TT pair (< IPR > = 2.5), while the hole is localized on one site and partly on the neighbor one (< IPR > = 1.4) as shown with magenta dots in Fig. 4c, d, respectively. These values becomes slightly smaller at 150 K for both holes and electrons. The more localized nature of the hole wavefunction is consistent with our experimental observation of a larger $B_{\text{rms}}$ for holes than for electrons, which suggests that electrons are spread over more nuclei than holes. We estimate that the ratio of the intra-chain delocalization of a hole particle wavefunction in DPPT-TT to that of an electron · quantified using the inverse participation ratio—is about 0.54. This is in line with the ratio of the root-mean-square fluctuations of the magnetic field reported in Table 1. We note, however, that a more quantitative agreement cannot be expected since $B_{\text{rms}}$ is also potentially affected by nuclei on neighboring chains, and these are not considered in our calculations. We also note that $B_{\text{rms}}$ are likely to be dominated by hyperfine interactions and not strongly affected by g-factor anisotropy (see Supplementary Note 10). Interestingly, by looking at the IPR distributions in Fig. 4c, d, we see that these are much broader for n-type carriers than for p-type carries at both 150 and 300 K. This means that while hole carriers are confined mostly to a single site, electrons can spread over a variable number of sites. This is a consequence of the fact that the strong electron-phonon interactions characterizing hole carriers give rise to the formation of deep-trap hole states at the top of the valence band. This is clearly shown in Fig. 5a, b, where we depict the delocalization of the valence band states (using the $IPR_i$ definition in eq. (10)) as a function of their energies). The formation of a third band of states—induced by local electron-phonon coupling and structural reorganization—at the top of the valence band is also shown in the DOS in Fig. 5d.

In Fig. 5f, by analyzing a representative surface-hopping trajectory, we see that the hole wavefunction spends most of its time in the

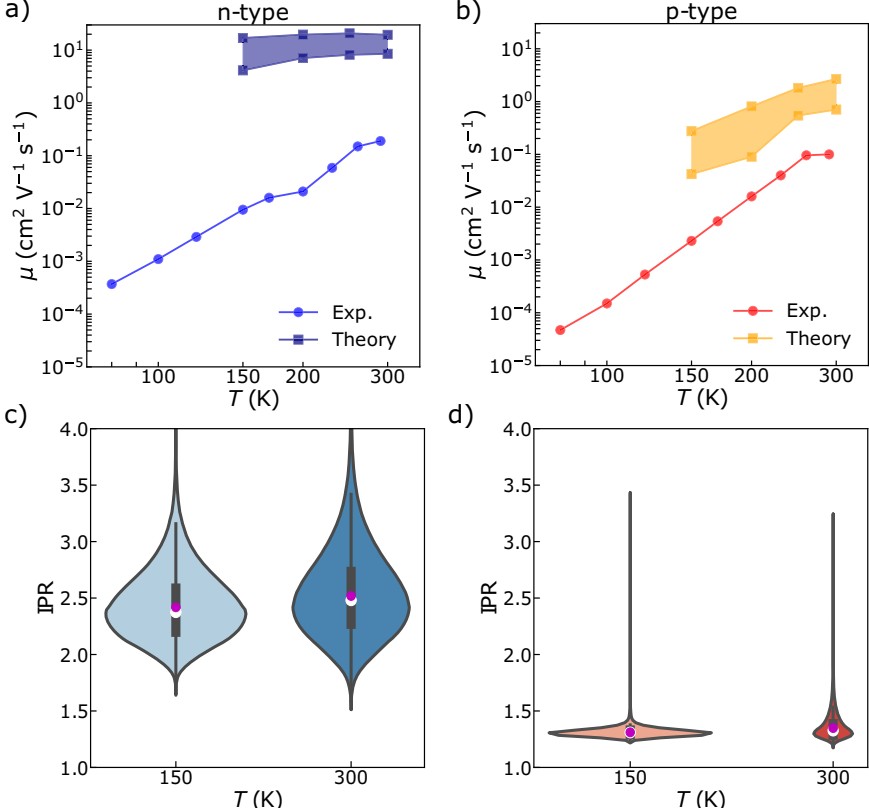

**Fig. 4 | Temperature-dependent experimental and theoretical hole and electron mobilities for DPPT-TT and related inverse participation ratio (IPR) distributions. a**, **b** Experimental polaron mobilities reproduced from Fig. 3d and corresponding theoretical mobilities calculated as described in the text for two different values of $\lambda_{ext}$ (i.e., 100 and 150 meV). The blue and orange shaded regions indicate the interval spanned by these two values. Theoretical intra-chain mobilities are given as an average over 10,000 surface-hopping trajectories. **c**, **d** Violin plots representing IPR distributions (obtained from the same surface-hopping trajectories) at 150 K and 300 K for electron and hole wavefunctions, respectively. In this case, $\lambda_{ext} = 150$ meV is shown. Black bars in the center represent interquartile ranges, while the thinner black lines stretching from the center represent Tukey's fences. White and magenta dots represent medians and means of the distributions, respectively. The mode of the distributions can be inferred by their maximum width. Note that electron IPR distributions have longer tails at all temperatures.

lowest energy state, i.e., at the top of the valence band and primarily localized on DPPT monomers, and it follows an essentially adiabatic time evolution (see the red line in Fig. 5f, which represents the active potential energy surface). Only rarely (and momentarily) does the hole access slightly higher-energy adiabatic states (i.e., states deeper within the valence band), and these states can be delocalized over two or more sites. The transient access to slightly more delocalized states occurs because of the continuous energy exchange with vibrational modes in the polymer, serving to kick the charge over to a nearest neighbor or even longer distances. This "transient delocalization" mechanism, which characterizes charge transport[29] (as well as exciton transport[30–32]) in other semiconducting materials[33] is encountered more frequently by the electron wavefunction. This becomes evident when looking at the skewness of the electron IPR distribution towards larger IPR values in Fig. 4c compared to one of the holes in Fig. 4d. The more effective transient delocalization of electrons compared to holes is due to their smaller local electron-phonon coupling and larger electron bandwidth creating delocalized, thermally accessible states close to the bottom of the conduction band (see Fig 5a). Such a mechanism also explains why electrons have larger mobilities than holes and why these mobilities increase with increasing temperature. In fact, we can observe that for both holes and electrons, the IPR distributions become slightly more skewed as the temperature increases. This is due to the larger thermal energy available, which favors a more efficient energy exchange between electrons and vibrational modes capable of triggering transient carrier delocalization

events, larger spreading of the charge density (as shown in Supplementary Fig. 8) and faster mobilities.

The vibrational dynamics of the polymer backbone cause displacements of the wavefunctions along the chain. This adiabatic motion along the backbone in response to torsion is the cause of spin relaxation in the spin-shuttling regime at high temperatures. These simulations show that electron and hole wavefunctions evolve in a similar manner on a picosecond timescale, but the electron wavefunction tends on average to be more delocalized and diffuse faster.

## Discussion

Based on the results of these simulations and our experimental data, we argue that spin-shuttling relaxation sets in later for holes because the hole wavefunction, which is inherently more localized, is less susceptible to the torsional events that drive transient delocalization and relaxation events. Particular modes of interest are those occurring across the TT-T bonds since the radical cation is localized primarily over a DPPT unit and has little amplitude beyond the adjacent TT unit (see also Supplementary Fig. 4). This should be compared to the electron wavefunction, which spreads from the central DPP unit over the nearest thiophene unit and onto the TT unit. From this, one would expect the electron wavefunction to be susceptible to torsional vibrations of the T-TT bonds, while the hole wavefunction is likely to be less susceptible. This provides an explanation of our observation above that the spin-shuttling regime sets in earlier and leads to shorter relaxation times for electrons, reversing the trend observed in the

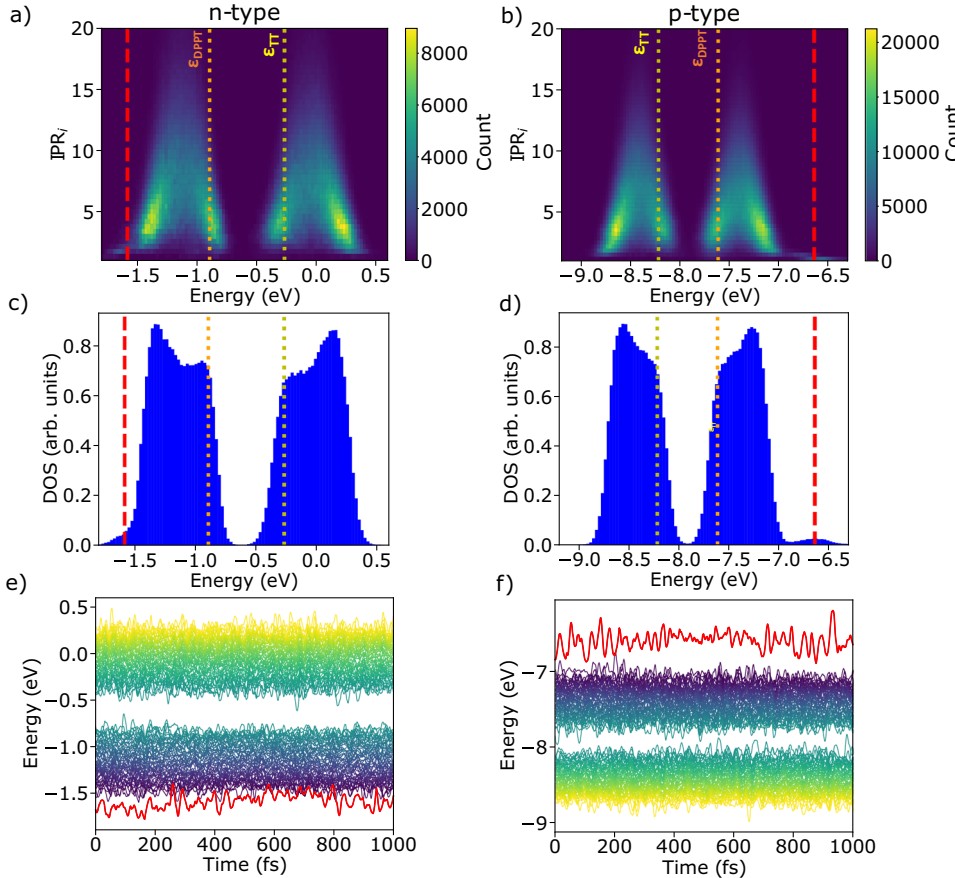

**Fig. 5 | State-resolved inverse participation ratio, density of states (DOS) and potential energy surfaces as a function of time. a**, **b** 2D histograms correlating the delocalization of the valence band and conduction band states, respectively, quantified by binning the inverse participation ratio IPR$_i$ of the adiabatic states (eq. (10)) versus their energies. The states become denser moving from dark blue, to green, to yellow regions. The color bar indicates the states count in a given bin. Vertical dashed red lines indicate the (average) active state energy (see Methods), while the dotted orange and yellow lines indicate the energies of the HOMOs and LUMOs of DPPT ($\epsilon_{DPPT}$) and TT ($\epsilon_{TT}$) units found from periodic DFT calculations at 0 K. **c**, **d** The DOS of conduction and valence band states of the system, respectively. DOS and state-resolved IPRs (IPR$_i$) are computed from Hamiltonians extracted from 10 surface hopping trajectories carried out at 300 K (including the effect of thermal disorder). Vertical lines continue from the upper panels (**e**) and (**f**) The time-evolution of the active state (red) and the corresponding full eigenspectrum along a representative trajectory for the polymer system in the case of electrons and holes.

motional narrowing regime (in which the higher electron mobility facilitates weaker spin relaxation). We hypothesize, therefore, that in this polymer spin relaxation in the spin-shuttling regime is driven primarily by the coupling of charges to the torsional motion of the T-TT bonds.

In general, the advantage of our experimental technique is that we are able to make comparisons between electron and hole polarons in the same polymer microstructure and the same structural dynamics. By comparing electron and hole wavefunctions in our two fused systems (for which polaron wavefunctions are expected to be more similar to each other) to those in the donor-acceptor systems (where polaron wavefunctions are expected to differ more from one another), we here have been able to clarify the extent to which different relaxation regimes are modulated by wavefunction localization and by charge-transport characteristics.

In all three systems, we observed for both electrons and holes the three regimes of spin relaxation previously reported in conjugated polymers: inhomogenous broadening, motional narrowing, and high-temperature torsion-induced spin-shuttling relaxation. This provides further evidence for the generality of the observations reported in ref. 18. There are clear differences between electrons and holes in both the onsets and the magnitude of the spin relaxation times in the different regimes. In the motional narrowing regime the spin relaxation behavior is closely correlated with the measured FET

mobility. In the two fused systems (AN and NN), motional narrowing sets in at the same temperature for both holes and electrons, while in DPPT-TT electrons display a motionally narrowed signal at much lower temperatures than holes. This observation is readily understood by considering the differences in polaronic mobilities: in the fused systems, mobilities are nearly equal at low temperatures (thus meaning motional narrowing sets in for both at the same temperature), while in DPPT-TT there is over an order-of-magnitude difference in electron and hole mobility below 150 K. In all three polymers electrons exhibit longer spin relaxation times than holes in this regime, which is consistent with their mobility being higher than that of holes.

Moving away from motional narrowing, all three systems show a difference between holes and electrons in the onset of high-temperature relaxation, which reverses the trend observed in the motional narrowing regime. The spin-shuttling mechanism sets in earlier for electrons by approximately 20 K in the fused systems and 50 K in DPPT-TT, and electrons tend to exhibit a shorter spin relaxation time than holes at high temperatures. Through an estimation of the delocalization length of holes vs. electrons via hyperfine broadening, (and which is consistent with electronic structure simulations), we attribute these different onsets to the spread of polaronic wavefunctions. Holes are not only more localized than electrons, but are also localized away from the relevant sites of torsions. They are thus less

affected by torsion than electrons, and therefore higher temperatures (resulting in higher torsional amplitudes) are needed to effect the same magnitude of relaxation.

Our results show that comparisons of electron and hole spin relaxation in the spin-shuttling regime provide an intimate probe of vibration-induced wavefunction dynamics in conjugated polymers and give insight into the unique and important coupling between structural, charge, and spin dynamics. We have established that more mobile electron carriers with extended wavefunctions experience stronger spin relaxation in the spin shuttling regime than less mobile hole carriers with more localized wavefunctions. A full quantitative, theoretical model is still needed that is able to directly simulate how the vibrational dynamics and shuttling motion of the electron wavefunction mediate spin relaxation, but ambipolar FI-ESR provides a direct and broadly applicable experimental probe of these microscopic processes that govern the charge transport physics of these complex materials.

## Methods

### FET preparation

Field-induced ESR measurements were performed on top-gate, bottom-contact FET devices: Fused-quartz plates (UQG Optics, FQP-5005) were cut to $40 \times 40$ mm squares with a diamond saw, then ridges of depth 0.3 mm were cut into one side of the glass along its full length at the 3 mm mark, 4.5 mm mark, 7.5 mm mark, 9 mm mark, etc. After cleaning, we used photolithography followed by metal evaporation to deposit interdigitated source and drain contacts with a total channel width and length of 243 mm and 0.1 mm, respectively. The odd layout of the electrodes is due to the narrow constraints of the ESR cavity coupled with the need for high carrier injection in order to detect an ESR signal.

Solutions of AN and NN were prepared by adding 10 g/l of polymer to trichlorobenzene and dissolving for 1 h at 160 °C. We heated the substrates at 150 °C for 5 min, then spin-coated the solutions onto them at 1200 rpm for 6 min using hot glass pipettes (also heated at 150 °C for 5 min). Immediately afterward we annealed the devices by placing them on a hot plate at 160 °C for 5 min, then 250° for 30 min, then quenching them. It should be noted that this fabrication procedure differs slightly from the one reported by our group in ref. 24; this was done intentionally in order to achieve better ambipolar transport.

The solution of DPPT-TT was created by adding 10 g/l of material to dichlorobenzene and dissolving overnight at 80 °C. Solutions were spin-coated at 1400 rpm for 60 s. Devices were then annealed at 320 °C for 20 min, then quenched.

After polymer deposition, all devices had PMMA spin-coated onto them at 1400 rpm for 30 s, then were annealed at 80 °C for 30 min. This method results in a PMMA layer that is 400 nm thick and has a dielectric constant of 3.6. PMMA is known to be suitable for ambipolar transport in polymer FETs because it has few electron-trapping groups[5,34–36].

We finished fabrication by evaporating aluminium gates of thickness 30 nm over the active areas. Individual devices were separated from the $4 \times 4$ cm square by applying pressure along the length of the 0.3 mm indents. In some cases, it was helpful to first cut through the dielectric and polymer over the grooves using a scalpel. This prevented the dielectric from peeling off the gate upon device separation.

### FI-ESR measurements

To perform FI-ESR measurements, a transistor was attached and wire-bonded to a substrate holder with source, drain, and gate connections. The device-and-boat combination was lowered into a tube appropriate for ESR measurements and sealed under nitrogen using a rubber cap. The electrode wires were punctured through the cap in order to connect to the voltage source, and the puncture sites were sealed with epoxy to preserve vacuum.

All EPR measurements were taken on a Bruker E500 spectrometer using a Bruker ER 4122SHQE cavity and an X-band microwave source. An Oxford Instruments ESR900 cryostat controlled by an Oxford Instruments Mercury iTC was used for temperature-dependent spectra, and a Keithley 2602b source unit was used for electrical characterization. CustomXepr, a Python package developed by ref. 18, was used to integrate data collected by these instruments and to automate measurements when desired.

### Mixed quantum-classical dynamics

As mentioned above, to study intra-chain hole and electron transport in DPPT-TT we used the advanced crossing-corrected variant[26,27] of Tully's fewest switches surface hopping algorithm[28], which incorporates non-adiabatic transitions between different adiabatic potential energy surfaces (PESs), i.e., states within the conduction or valence bands in our system. This technique allows one to accurately propagate coupled electron-nuclear motion in long polymer chains thanks to novel algorithmic improvements that efficiently deal with complex surface crossings[37]. In surface hopping, the dynamics of the system are described by an ensemble of independent trajectories, where each trajectory occupies an "active" PES at individual time steps. The electron wavefunction $|\Psi(t)\rangle$ can be expressed as a linear combination of the eigenstates of the Hamiltonian (see eq. 10 in Supplementary Note 7),

$$|\Psi(t)\rangle = \sum_i w_i(t)|\phi_i(\mathbf{r}; \mathbf{R}(t))\rangle, \tag{3}$$

where $w_i(t)$ are expansion coefficients and $\mathbf{R}$ represents the nuclear degrees of freedom. The carrier wavefunction is propagated along each trajectory by solving the time-dependent Schrödinger equation (TDSE),

$$\frac{\mathrm{d}w_i}{\mathrm{d}t} = \frac{w_i(t)E_i(\mathbf{R}(t))}{i\hbar} - \sum_j w_j(t)\frac{\mathrm{d}\mathbf{R}(t)}{\mathrm{d}t}\mathbf{d}_{ij}(\mathbf{R}(t)) \tag{4}$$

where $\mathbf{d}_{ij}(\mathbf{R}(t))$ is the non-adiabatic coupling vector. Nuclear dynamics on the active PES are modeled by the Langevin equation[38],

$$M\frac{\mathrm{d}^2\mathbf{R}}{\mathrm{d}t^2} = -V' - \gamma M\frac{\mathrm{d}\mathbf{R}}{\mathrm{d}t} + \zeta, \tag{5}$$

where $-V'$ is the effective force on the active PES, $M$ the mass equivalent corresponding to the classical nuclear degrees of freedom, $\gamma$ is the friction coefficient (set to 100 ps$^{-1}$, as in previous works[26]), and $\zeta$ is a Markovian Gaussian random force (for details, see ref. 38). To account for the feedback of nuclear motion on the electronic motion and to achieve internal consistency between electronic population of the states and the fraction of trajectories on each PES[39], stochastic hops between adiabatic PESs were implemented as described in Tully's approach[28]. Without going into much detail, the method employed in this work features important improvements over the original surface hopping approach, as necessary for robust and meaningful mobility calculations[37,40]. These algorithms (which are reviewed in detail elsewhere[26,27,37,40,41]) include the following: (1) a decoherence correction, which in this work is implemented using the energy-based decoherence time[25,26] (neglecting the size-extensive kinetic energy term, as done in ref. 40); (2) a re-scaling of the velocities upon successful hops along the non-adiabatic coupling vectors to reach the correct detailed balance of the states[37,41]; and (3) a novel trivial-crossings detection algorithm which allows one to carry out dynamics in a dense manifold of adiabatic states[26,27]. In this work, we model polymer chains as one-dimensional arrays with $N = 101$ sites (employing open boundary conditions). The nuclear time-step size was set to $dt = 0.01$ fs and simulations were carried out for at least 1 ps.

## Mobility calculations and IPRs

Solving the TDSE in eq. (4) allows one to obtain the charge-carrier wavefunction as a function of time, $|\Psi(t)\rangle$. This gives access to key dynamic properties, such as the mean squared displacement (MSD) and the extent of (de)localization of the charge as a function of time, as well as the mechanism by which the charge carrier moves within the polymer chain. The time-dependent MSD is calculated in this work using the charge-carrier wavefunction of each trajectory $n$, $|\Psi_n(t)\rangle$, as

$$\text{MSD}(t) = \frac{1}{N_{\text{traj}}} \sum_{n=1}^{N_{\text{traj}}} \langle \Psi_n(t) | (r - r_0)^2 | \Psi_n(t) \rangle, \qquad (6)$$

where $N_{\text{traj}}$ is the number of independent trajectories, which here is set to 10,000 to obtain a smooth and linear time-evolution profile of MSD(t); $r_0$ is the average position of the charge carrier. The matrix elements in the above equation are computed exploiting the fact that the charge-carrier wavefunction can also be represented in the monomer-based diabatic representation as

$$\Psi(t) = \sum_{k}^{N} c_k(t) |k\rangle, \qquad (7)$$

where $c_k(t)$ are the diabatic expansion coefficients on different sites $k$. In this basis, since we are dealing with a single polymer chain, $\langle k|r^2|l\rangle = \delta_{kl} k^2 L^2$ and $\langle k|r|l\rangle = \delta_{kl} kL$, $L$ being the distance between nearest-neighbor monomer components. Since the charge is initially placed at the central monomer of the chain, the reference position of the charge is $r_0 = NL/2$.

A linear evolution of the MSD(t) signifies that an equilibrium diffusion regime is attained, and the charge mobility can then be computed from the Einstein relation

$$\mu = \frac{qD}{k_{\text{B}} T}, \qquad (8)$$

where $D = \frac{1}{2} \lim_{t \to \infty} \frac{d\text{MSD}}{dt}$. We can also track the delocalization of the charge along the polymer chain using the time-dependent inverse participation ratio (IPR). This quantity gives a measure of how many sites the hole (or electron) wavefunction is delocalized over as a function of time; it is computed as

$$\text{IPR}(t) = \frac{1}{N_{\text{traj}}} \sum_{n=1}^{N_{\text{traj}}} \frac{1}{\sum_{k}^{N} |c_{k,n}(t)|^4}. \qquad (9)$$

Such a quantity can be average over time to obtain $\langle \text{IPR} \rangle$, that is the average extension of the carrier wavefunction. The delocalization of the eigenstates of the Hamiltonian of the system can be found using the IPR of a given adiabatic state $i$ at a given time $t$ in a specific trajectory:

$$\text{IPR}_i(t) = \frac{1}{\sum_{k}^{N} |\langle k|\phi_i(t)\rangle|^4}. \qquad (10)$$

This quantity is binned over time and trajectories as a function of energy to produce Fig. 5a, b.

## Data availability

The datasets generated during and/or analyzed during the current study are available in the Zenodo repository at https://doi.org/10.5281/zenodo.10026980. The full data for this study totals a few hundred of Gigabytes, so it is in cold storage accessible by the corresponding author and available upon request.

## Code availability

The CustomXepr source code is available at https://github.com/OE-FET/customxepr and other software used is available upon request.

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

## Acknowledgements

We acknowledge funding from the European Research Council through a Synergy grant (610115). H.S. also acknowledges support by a Royal Society Research Professorship (RP\R1\201082) and an ERC Advanced grant (101020872). The work in Mons received funding from the European Union's Horizon 2020 research and innovation program under grant agreement No. 964677, the Consortium des Équipements de Calcul Intensif (CÉCI), funded by the Fonds National de la Recherche Scientifique (F.R.S.-FNRS) under Grant No. 2.5020.11 as well as the Tier-1 supercomputer of the Fédération Wallonie-Bruxelles, infrastructure funded by the Walloon Region under Grant Agreement n1117545, and F.R.S.-FNRS. S.G. is Chargé de recherches FNRS, C.Q. is FNRS research associate and D.B. is FNRS Research Director. L.W. acknowledges support from the National Natural Science Foundation of China (Grant No. 22273082). S.P. acknowledges financial support from the European Research Council (Grant No. 101020369).

## Author contributions

R.L.C. and S.S. conceived the experiments, R.L.C. and M.X. fabricated the devices, and R.L.C. and S.S. conducted the experiments. S.G. parameterized the DPPT-TT coarse grain model, extended the surface hopping code to deal with the specific problem and performed nonadiabatic dynamics (with inputs from S.P.). V.L., M.B., and C.Q. performed the quantum-chemical and periodic DFT calculations. L.W. provided the original code for surface hopping simulation of charge transport and joined in the discussions. R.L.C., S.G., S.S., V.L., D.B., and H.S. analyzed the results and wrote the manuscript. All authors reviewed the manuscript.

## Competing interests

The authors declare no competing interests.
