## [Peer Review File · Nature Communications]

REVIEWER COMMENTS

Reviewer #1 (Remarks to the Author):

The manuscript "Spin Relaxation of Electron and Hole Polarons in Ambipolar Conjugated Polymers" by Carey et al. is a detailed, extensive report that uses electron spin resonance to investigate the spin relaxation of polarons in ambipolar semiconducting polymers, the first time now on both the electron and the hole side. The detailed experiments are connected to equally detailed theory. The work extends the beautiful work by the same group of authors reported in 2019 (Schott et al Nat. Phys), where they had shown that by temperature dependent ESR measurements they can identify different spin relaxation regimes and draw from these conclusions about the delocalization of wavefunctions and their coupling to the nuclei. This current report now builds on this pioneering work in that it tests the different spin relaxation regimes in a set of three ambipolar polymers. The work is done with great care and well-written. I think the manuscript could be suitable for Nat Com after a major revision if the following points are addressed:

1.) The main claim of the paper and the advancement compared to the group's prior work is, that differences in hole and electron wavefunction dynamics are visualized by ESR. This general statement however is very similar to the result of the Nat Phys 2019 result. In general, is very much expected that due to the different local distribution of the HOMO and LUMO orbitals, also the coupling between charges and the polymer backbone will be different. In general, I find that the systematic difference between hole and electron spin relaxation in comparison to the respective charge carrier mobility does not significantly differ when comparing the respective relaxation times and mobilities between the polymers of one charge carrier type.

2.) Extraction of charge carrier mobilities (also connects to point 1): the charge carrier mobilities are heavily gate dependent. What is the error bar in extracting the mobilities at constant gate voltage? How do the conclusions change when they perform measurements at a different charge carrier mobility? Given that the threshold and onset voltages of transistors fabricated from the three polymers are different, the experiments should be performed at the same accumulated (electron and hole) charge carrier density, and not at the same gate-source voltage. Also, I find that the statement in the caption to Figure 3 "At higher temperatures, polaronic mobilities are nearly equal in DPPT-TT, while in AN and NN they diverge." is not reflected by the data – it seems that that in AN and NN the electron and hole mobilities become more similar rather than diverging.

Minor points:

- In the first sentence of the introduction, the authors state that "an ideal semiconductor" should transport both electrons and holes with equal mobility. I think in general I would not agree to this statement – there are applications where unipolar transport is beneficial, others where ambipolar transport is beneficial.

- Page 4, middle of last paragraph: "if deemed necessary.." background signal was measured – could the authors add some qualification to this statement?

Reviewer #2 (Remarks to the Author):

The paper by Carey et al. studies origins of the spin relaxations of the organic semiconductors at several different temperatures using field induced electron spin resonance spectroscopy. In the motional narrowing regime between 100 and 200K electron spin relaxation is weaker than for those in holes, reflecting a higher electron mobility. In the spin-shuttling regime above 200K this trend reverses and electrons experience stronger spin relaxation. On the basis of theoretical simulations they attribute this to differences in the delocalisation of electron and hole wavefunctions and show that spin relaxation in the spin shuttling regimes provides a sensitive probe of the intimate coupling between charge and structural dynamics. This is an interesting

contribution on the origins of the carrier mobilities in connection to the chemical structure and to the electronic orbital characteristics. However, this reviewer raises some concerns about the present interpretations:

- 1) Low temperature ESR: The authors only present the ESR spectra at 200 K wherein the motional narrowing effects seem to be significant. Please show some of the corresponding ESR spectra to the readers at 10 K and 100 K. Did you see the anisotropies in the g-factors and in the hyperfine structures at low temperature?
- 2) Method to obtain T1 and T2: How did you obtain the relaxation parameters in Figure 3 from the "power-saturation measurements"? B1 dependences of the EPR intensities are well fit by the theory? More explicit descriptions and data are required for the readers. Furthermore, how did you discriminate between T1 and T2 in the present analysis?
- 3) Motional Narrowing: The extracted hyperfine fields in Table 1 are explained by the longer delocalization length in the electron than in the hole, associated with the charge mobilities. Can you rationalize the extracted hyperfine fields in Table 1 using the distribution in Supplementary Figure 3?
- 4) Concerning the mobility, this could be limited by some minor defect sites in the polymer molecules during the charge hopping dynamics, while the present hyperfine field strength seems to play a major role on the local charge-conducting character. Please comment on this possible mismatch.
- 5) Spin Shuttling: It is of great interest to consider the effects of the spin shuttling on the spin relaxation at the elevated temperatures. At page 10, the authors mention that the torsional vibrations play a role on presenting the opportunity for the spin relaxations. However, such torsional vibrations themselves can allow the g-anisotropy and the hyperfine anisotropy to be averaged contributing to the motional narrowing effect. Thus, do the torsional vibrations contribute also to the quicker spin relaxation in TPPT-TT around 100 K regions? Some comments are required because the torsional vibrations are possible to occur at 100 K.
- 6) In this regard (items 3 and 5), the anisotropies in the g-factor and in the hyperfine coupling may contribute to the extracted hyperfine fields together with the delocalization lengths. Please present the anisotropies in the EPR parameters possibly obtained by the DFT calculations for the electrons and holes in DPPT-TT, AN and NN. Then, please discuss these effects on the narrowing and on the broadening with presenting the frequency of the vibrations.
- 7) The simulations in Figure 4: What are the causes in the discrepancies between the theory and experiment? See item 4.

Overall this paper could be acceptable for the publication if the authors address all the above concerns for the quality of the present work.

Response Letter

Dear Editor,

Enclosed is a revised manuscript submission entitled:

“Spin Relaxation of Electron and Hole Polarons in Ambipolar Conjugated Polymers”

by Carey, R. L., et al. for publication in Nature Communications.

We thank you for your careful handling of our manuscript and the reviewers for their helpful and constructive comments, which have helped us to improve our manuscript. Below we give a point-by-point response in blue text to the questions raised and a description of the changes in the manuscript. A pdf file of the manuscript with all changes indicated in red text and a related Supplementary Information file have been submitted as supporting material.

Reviewer 1

The manuscript “Spin Relaxation of Electron and Hole Polarons in Ambipolar Conjugated Polymers” by Carey et al. is a detailed, extensive report that uses electron spin resonance to investigate the spin relaxation of polarons in ambipolar semiconducting polymers, the first time now on both the electron and the hole side. The detailed experiments are connected to equally detailed theory. The work extends the beautiful work by the same group of authors reported in 2019 (Schott et al Nat. Phys), where they had shown that by temperature dependent ESR measurements they can identify different spin relaxation regimes and draw from these conclusions about the delocalization of wavefunctions and their coupling to the nuclei. This current report now builds on this pioneering work in that it tests the different spin relaxation regimes in a set of three ambipolar polymers. The work is done with great care and well-written. I think the manuscript could be suitable for Nat Com after a major revision if the following points are addressed:

We thank the reviewer for their positive assessment of our work and for suggesting that it might be suitable for publication in Nature Communications. Below we address their concerns.

Concern 1

Concern: The main claim of the paper and the advancement compared to the group’s prior work is, that differences in hole and electron wavefunction dynamics are visualized by ESR. This general statement however is very similar to the result of the Nat Phys 2019 result. In general, is very much expected that due to the different local distribution of the HOMO and LUMO orbitals, also the coupling between charges and the polymer backbone will be different. In general, I find that the systematic difference between hole and electron spin relaxation in comparison to the respective charge carrier mobility does not significantly differ when comparing the respective relaxation times and mobilities between the polymers of one charge carrier type.

Response: Our previous Nature Physics (Schott et al., 2019) proposed a hypothesis for the unusual high-temperature spin relaxation behavior of hole polarons in conjugated polymers. The research presented in this current work provides a further stringent test of the so-called spin

shuttling mechanism proposed in the Nature Physics paper. In that paper, we had no way to test the significant influence of the carrier wavefunction on spin-shuttling relaxation because comparisons between hole polarons in different polymers involve too many confounding factors. In our present work we show for the first time how to remove these confounding factors. For three different materials, we show that comparisons between electron and hole polarons in the same polymer system allow one to obtain information on how the degree of localization of the charge wavefunction along the polymer backbone affects spin relaxation behavior. We show that we are able to consistently explain the observed differences between electron and hole spin relaxation within the spin shuttling framework. This provides an important validation of the mechanism proposed by our previous work and moves it closer to becoming an accepted framework for interpreting the spin relaxation behavior of this important class of functional materials. We believe that this is a methodical application of the scientific method: a mechanism for a phenomenon is proposed, then validated by future work, and finally becomes accepted (or invalidated by more careful experiments).

We have further clarified this at the end of paragraph 4, page 3, in the Introduction section of the paper: “Unfortunately, we were not able to test this dependence on wavefunction dynamics in our previous paper at that time because our study relied on comparisons between polymers, but such comparisons also naturally include variation in system microstructure and vibrational dynamics, both of which have confounding effects on spin relaxation.”

In response to the referee’s comment on the motional narrowing behavior of polarons: The relationship between mobility/motion frequency and spin lifetimes is indeed the same when comparing holes to electrons and comparing holes across different unipolar systems. This is because the motional narrowing mechanism is not sensitive to the localisation of the wavefunction (other than through the mobility/motion frequency). We included the motional narrowing data here to show that the electron and hole spin relaxation times do indeed behave as expected from the respective mobilities and also to highlight the transition to the scientifically more interesting/novel high-temperature relaxation regime.

Concern 2

Concern: Extraction of charge carrier mobilities (also connects to point 1): the charge carrier mobilities are heavily gate dependent. What is the error bar in extracting the mobilities at constant gate voltage? How do the conclusions change when they perform measurements at a different charge carrier mobility? Given that the threshold and onset voltages of transistors fabricated from the three polymers are different, the experiments should be performed at the same accumulated (electron and hole) charge carrier density, and not at the same gate-source voltage. Also, I find that the statement in the caption to Figure 3 “At higher temperatures, polaronic mobilities are nearly equal in DPPT-TT, while in AN and NN they diverge.” is not reflected by the data – it seems that that in AN and NN the electron and hole mobilities become more similar rather than diverging.

Response: We agree with the referee that it is important in principle to take into account differences in threshold/onset voltages when comparing mobilities for electrons and holes.

However, despite the gate-voltage-dependent mobility, the device characteristics of our ambipolar devices (in particular DPPT-TT and AN) are very well-behaved and the onset voltages are very close, typically within ± 5 V to 0 V for both electrons and holes. Since we measured the ESR spectra at high magnitudes of gate voltage — ± 60 V — the associated uncertainty in the mobility is estimated to be less than 10%. This uncertainty is much smaller than the difference between electron and hole mobility at a fixed $\pm V_g$ (which is a factor of approximately 2 in the case of DPPT-TT, note the logarithmic scale in Figure 3 d-f). Therefore, given that the origin of the relatively small variations in onset voltage is not known, we felt that it was more transparent to compare electron and hole ESR spectra at the same total induced charge concentration and not correct for onset voltage.

We have added almost verbatim the red portion of the preceding paragraph to page 5 of the manuscript.

It is also worth noting that the gate-voltage-dependent mobility does not show any obvious artifacts of mobility extraction (such as peaks at relatively low gate voltages), which typically lead to mobility overestimation. The mobility increases gradually with gate voltage, a fact likely to reflect genuine effects of the energetic disorder in the density of states, which means that high mobility states only become populated once a distribution of shallow trap states has been filled. Our ESR measurements at high gate voltages probe transport in a regime where the majority of charges/spins are in such mobile states.

Finally, we agree with the Reviewer that the caption for Fig. 3 is not worded correctly. We have changed it accordingly in the main text.

Concern 3

Concern: In the first sentence of the introduction, the authors state that “an ideal semiconductor” should transport both electrons and holes with equal mobility. I think in general I would not agree to this statement – there are applications where unipolar transport is beneficial, others where ambipolar transport is beneficial.

Response: We agree that this is not correctly worded and we have changed the opening statement in the first paragraph of the introduction to the following: One of the characteristics of a clean, ambipolar semiconductor is the ability to support with comparable carrier mobilities the transport of both electrons in the conduction band and holes in the valence band. Any differences between the electron and hole mobilities should reflect differences in the intrinsic electronic structure of the conduction and valence band states, but should not be the consequence of defect states in the band gap that might trap one of the two carriers more strongly than the other. Of course, unipolar semiconductors, in which one type of charge is significantly more mobile than the other, can be very useful for many applications, but ambipolar semiconductors are particularly interesting for fundamental studies because they allow one to compare the charge-transport properties of carriers in the valence and conduction bands.

Concern 4

Concern: Page 4, middle of last paragraph: “if deemed necessary..” background signal was measured – could the authors add some qualification to this statement?

Response: We agree that this should be clarified. We have created Supplementary Note 8 and added the following clarification to it: In ESR studies, the test tubes used for measurement have extremely low amounts of magnetic impurities to prevent spurious contributions to the ESR signal. However, even small amounts of impurities can be detected at low temperatures due to the Curie susceptibility of such materials. We always use the highest-purity tubes available for our ESR studies, but occasionally even those have detectable levels of impurities at low temperatures. In such cases, the spurious signal can be removed by measuring our FI-ESR devices at nonzero gate voltage (i.e., under the desired measurement conditions) and subtracting from that signal a signal obtained by measuring at $V_g = 0$ V (which is the background signal). We followed this procedure here. We note that care must be taken in doing so because the microwave frequency may change between measurements, meaning the resonance condition equation must be used to properly align the two spectra.

Reviewer 2

The paper by Carely et al. studies origins of the spin relaxations of the organic semiconductors at several different temperatures using field induced electron spin resonance spectroscopy. In the motional narrowing regime between 100 and 200K electron spin relaxation is weaker than for those in holes, reflecting a higher electron mobility. In the spin-shuttling regime above 200K this trend reverses and electrons experience stronger spin relaxation. On the basis of theoretical simulations they attribute this to differences in the delocalisation of electron and hole wavefunctions and show that spin relaxation in the spin shuttling regimes provides a sensitive probe of the intimate coupling between charge and structural dynamics. This is an interesting contribution on the origins of the carrier mobilities in connection to the chemical structure and to the electronic orbital characteristics.

We thank the reviewer for their positive statement, and we address their important concerns below.

Concern 1

Concern: Low temperature ESR: The authors only present the ESR spectra at 200 K wherein the motional narrowing effects seem to be significant. Please show some of the corresponding ESR spectra to the readers at 10 K and 100 K. Did you see the anisotropies in the g-factors and in the hyperfine structures at low temperature?

Response: As briefly mentioned on page 8 of the original manuscript, all ESR spectra remained Lorentzian down to 5 K. This implies that no g-factor anisotropy nor hyperfine coupling could be resolved. To make this point clear, we have added several representative low-temperature spectra to Supplementary Note 10.

Concern 2

Concern: Method to obtain T1 and T2: How did you obtain the relaxation parameters in Figure 3 from the "power-saturation measurements"? B1 dependences of the EPR intensities are well fit by the theory? More explicit descriptions and data are required for the readers. Furthermore, how did you discriminate between T1 and T2 in the present analysis?

Response: The method for extraction of T1 and T2 is the same as in our 2019 Nature Physics paper, as briefly mentioned in the main text. However, to better explain this and make the paper self-contained, we have added a more detailed description of the method to a new Supplementary Note 9. That note derives and describes the mathematical function to which we fit the data in detail.

In summary, power-saturation measurements probe the ESR signal as a function of both magnetic field and microwave power. In a standard ESR experiment, where only the magnetic field is swept, the width of the resonance signal is determined by the spread of different resonance positions among the spin ensemble, a quantity characterized by T2; these measurements therefore give an estimate for the transverse relaxation times. Power-saturation measurements examine how the signal evolves with increasing microwave power. At high powers, the number of spin-flip-inducing photons per unit time is so high that the spin ensemble reaches equilibrium (having an equal number of spins in the two states) and is unable to relax between field scans; this results in no signal observed at resonance since the two populations are equal. Because relaxation-from-equilibrium is described by T1, these experiments provide an estimate for the longitudinal relaxation time. We therefore fit the full 2D dependence of the resonance signal on field position and microwave power to extract T1 and T2 unambiguously.

Concern 3

Concern: Motional Narrowing: The extracted hyperfine fields in Table 1 are explained by the longer delocalization length in the electron than in the hole, associated with the charge mobilities. Can you rationalize the extracted hyperfine fields in Table 1 using the distribution in Supplementary Figure 3?

Response: We thank the reviewer for this point. As argued below in response to Concern 7, it is not possible to predict the absolute value of Brms directly from the simulations in Supplementary Figure 3, but we can use the expected scaling relationship of Brms with $1/\sqrt{N}$ — where N is the number of nuclei that interact with the electron spin wavefunction — to estimate the ratio of N_hole/N_electron to be 0.68. This value is reasonably close to the calculated ratio of IPR_hole/IPR_electron = 0.54.

A discussion of this has been added to page 9 after the discussion of Brms values in Table 1: As will be discussed below, this conclusion is supported by quantum-chemical calculations in DPPT-TT. We estimate that the ratio of the intra-chain delocalization of a hole particle wavefunction in

DPPT-TT to that of an electron — quantified using the inverse participation ratio (Eq. 9) — is about 0.54. This is in line with the ratio of the root-mean-square fluctuations of the magnetic field reported in Table 1. We note, however, that a more quantitative agreement cannot be expected since B_{rms} is also potentially affected by nuclei on neighboring chains, and these are not considered in our calculations.

Concern 4

Concern: Concerning the mobility, this could be limited by some minor defect sites in the polymer molecules during the charge hopping dynamics, while the present hyperfine field strength seems to play a major role on the local charge-conducting character. Please comment on this possible mismatch.

Response: It is correct that, in principle, mobility can be severely limited by specific defect configurations that are not representative of the spin environment probed by ESR experiments. However, in the relatively high-mobility polymer systems we are investigating here, we observe generally a good correlation between the motional narrowing physics observed in ESR and the measured device mobilities --- a trend also shown in our previous work for other high-mobility polymer systems[18]. Also, in the present work, the observed correlation in the motional narrowing regime between the more effective spin relaxation of electrons and the higher electron mobility is evidence that transport in these systems occurs in a regime in which the configurations probed by ESR *are* representative of the environments that determine the charge mobility. This is probably a consequence of the relatively high mobility of the polymers investigated, where specific, “minor” defect sites acting as deep charge traps do not play an important role.

The preceding text in red has been added to the end of the Relaxation Dynamics section on page 10. We also refer to Concern 7 below for further discussion on this point.

Concern 5

Concern: Spin Shuttling: It is of great interest to consider the effects of the spin shuttling on the spin relaxation at the elevated temperatures. At page 10, the authors mention that the torsional vibrations play a role on presenting the opportunity for the spin relaxations. However, such torsional vibrations themselves can allow the g -anisotropy and the hyperfine anisotropy to be averaged contributing to the motional narrowing effect. Thus, do the torsional vibrations contribute also to the quicker spin relaxation in TPPT-TT around 100 K regions? Some comments are required because the torsional vibrations are possible to occur at 100 K.

Response:

We agree that this is a valuable point to clarify. It is true that low-energy torsional modes will be activated at temperatures lower than the onset of the spin-shuttling regime, though the shift from motional narrowing to spin shuttling should be understood not as a sudden activation of a specific vibrational mode that triggers spin shuttling, but as the temperature at which the weakening of spin relaxation from motional narrowing allows the dominant relaxation mechanism to be spin shuttling. This view is supported by our simulations, which reveal a gradual increase of wavefunction transient delocalization as temperature increases (see increasing skewness of the

IPR distribution with temperature, Fig. 4c,d), despite the fact that vibrational modes (and their specific frequencies) included in our model are the same at all temperatures. In particular, at room temperature, not only do we see a slightly larger delocalization of the wavefunction than at lower temperatures, but also a more skewed distribution of the IPR. As mentioned on page 14, this points to a more effective transient delocalization mechanism triggered by a more intense energy exchange between the charge and nuclear vibrations when the thermal energy is higher.

To clarify this, we have added the following information to page 11 after introducing the SSH-type Hamiltonian used in this work: In particular, as described in Supplementary Note 3, the dynamics of the nuclei are described by three effective, harmonic vibrational modes. One is a high-frequency (~ 1000 cm $^{-1}$) intra-monomer mode, while the other two are low frequency vibrations (< 40 cm $^{-1}$) coupled to an "external" mode (intended as an additional contribution from the environment to the reorganization energy, as further described below), and to an inter-monomer torsional mode between successive monomer units along the polymer axis. It is worth noting that these soft degrees of freedom are active even at low temperature and, in principle, they also tend to reduce spin lifetimes at lower temperature, but in a less effective manner than at high temperature. We note in passing that the transition between motional narrowing and spin shuttling should not be interpreted as a transition at which suddenly a particular vibrational mode starts to be activated that leads to spin shuttling, but rather as the temperature at which the spin relaxation from motional narrowing becomes so weak that the relaxation induced by spin-shuttling becomes dominant.

Concern 6

Concern: In this regard (items 3 and 5), the anisotropies in the g-factor and in the hyperfine coupling may contribute to the extracted hyperfine fields together with the delocalization lengths. Please present the anisotropies in the EPR parameters possibly obtained by the DFT calculations for the electrons and holes in DPPT-TT, AN and NN. Then, please discuss these effects on the narrowing and on the broadening with presenting the frequency of the vibrations.

Response: We have added a reply to this helpful point to in Supplementary Note 11: The extracted RMS B-field values represent the average hyperfine coupling constant weighted by the square root of the number of molecules over which the wavefunction is spread. The hyperfine tensor can be formally expanded in orders of v/c , where v is the speed of the polaron and c is the speed of light. Doing so yields first-order contributions from the contact and dipolar terms (the usual hyperfine terms), and second-order contributions from spin-orbit interactions. Anisotropy in both couplings are therefore captured by our reported values, but of course we only report the average of these couplings. Because spin-orbit coupling is a second-order interaction, anisotropy in the g-factor will have a comparatively weak contribution to the extracted values; the dominant contributions will be those of the hyperfine interaction(s).

Unfortunately, modelling the hyperfine interactions realistically and comparing them to the experimental values of Brms would go significantly beyond the level of simulations currently in the manuscript. At present, the simulations are focused on the spin-shuttling regime, as spin shuttling is an intrachain process with only limited influence of interchain interactions. Therefore,

single chain calculations as currently presented are sufficient. However, for a realistic assessment of the Brms experienced in the motional narrowing regime a realistic model for the 3D microstructure and the charge transport within it would be required. Although this is possible in principle with state-of-the-art simulations, we argue that the main focus of the paper is on the spin shuttling regime and feel that such motional narrowing simulations would go significantly beyond the scope of the current paper. Furthermore, there is no evidence that the experimental values for Brms are inconsistent with the other observations: As discussed in the reply to concern 3, the observed ratio of the Brms values for electrons and holes is reasonably close to the value one would expect from the predicted IPR values of electrons and holes.

Concern 7

Concern: The simulations in Figure 4: What are the causes in the discrepancies between the theory and experiment? See item 4.

Response: As discussed on page 11, the main reason for the discrepancy between calculated and experimentally observed mobility values is that the calculations are performed for intrachain transport along the polymer backbone and do not consider the slower interchain hopping process, which is generally accepted to be the rate limiting step that governs the mobility observed in real devices, which are much longer than the polymer chain length. For this reason, the intrachain mobility simulations were not meant to provide a direct comparison with the experimental device mobility values. They are merely included here to provide an assessment of how much faster the local charge motion along the polymer backbone is than the slow interchain hopping step. This has been clarified in the discussion.

The general understanding of the transport physics of these relatively high-mobility conjugated polymers is that device mobility is limited by the slow interchain hopping processes that charges undergo when they come to the end of a chain or when they encounter a conjugation defect along the polymer backbone; these occur on a length scale on the order of 10 nm. The microstructure of these high-mobility systems is sufficiently uniform that it is not specific, *minor* defect sites and deep trap states that limit mobilities, but rather these typical interchain hopping processes that need to occur on a 10 nm length scale. This is ultimately the reason we observe such a clear correlation between spin relaxation times in the motional narrowing regime and experimentally observed mobilities. As shown in our previous paper, it is possible to use the experimentally determined motion frequencies in the motional narrowing regime to estimate the corresponding hopping distances by using the Einstein relationship. If we perform this analysis on the present DPPT-TT system, the estimated hopping distance at 150 K is on the order of 15 nm, consistent with the interpretation that it is a regular interchain hopping process and not a minor defect site that governs the mobilities and spin lifetimes in the motional narrowing regime.

Additional changes not requested by the reviewers.

We have slightly rephrased the abstract to improve its clarity. We have also slightly improved the method section and added a Code Availability statement in accordance with the Editorial Checklist

guidelines. We also have edited page 4 to show a statistical average for the anthracene-naphthalene copolymer's mobility from previous work.

Finally, we have altered the order of our supplemental notes so that they match the order in which they are now introduced in the main text. The mapping from old to new notes is as follows:

Old	New
	Note 1: NEW, as requested by reviewers
	Note 2: NEW, as requested by reviewers
	Note 3: NEW, as requested by reviewers
Note 1	Note 4
Note 2	Note 5
Note 3	Note 6
Note 4	Note 7
Note 5	Note 8
Note 6	Note 9
	Note 10: NEW, as requested by reviewers
Note 7	Note 11

We hope that all comments by the referees have been addressed duly and adequately and that the revised version of this paper can now be accepted for publication in *Nature Communications*.

REVIEWERS' COMMENTS

Reviewer #1 (Remarks to the Author):

The authors have greatly revised their work and I can now recommend it for publication.

Reviewer #2 (Remarks to the Author):

I am satisfied with the revision by the authors. Thus, the present version is worthy of publication.